# Zero-Shot Text-to-Motion Evaluation using Video Language Models

Yuwen Ji [1 2]   Donglin Wang [2]   Yue Zhang [2]

## Abstract

Text-to-motion (T2M) generation has become a fundamental task, yet existing evaluation metrics often fail to capture whether a generated motion semantically matches its text description. We propose VeMo, a zero-shot evaluation framework that renders generated human motions into videos and uses pretrained video-language models (VLMs) to assess text-motion alignment. Instead of training an evaluator on scarce motion-specific labels, VeMo transfers the semantic reasoning ability of VLMs to T2M evaluation through normalized likelihood-based scoring. To reduce the effect of 3D-to-2D projection ambiguity, we introduce an entropy-driven uncertainty analysis for identifying reliable rendered views. To address the lack of rigorous standards in the field, we further contribute a test-only and human-annotated meta-evaluation benchmark, covering motions generated by multiple representative T2M models. Extensive experiments show that VeMo correlates better with human judgments than existing reference-based and reference-free metrics. Theoretical and empirical analyses characterize both the promise and limitations of the VLM-based T2M evaluation. https://github.com/spatial-westlakenlp/ActionReward/tree/main/VeMo

## 1. Introduction

Text-to-motion (T2M) (Obludzyner et al., 2024) has emerged as a foundational setting for human motion generation. The objective of text-to-motion is to produce continuous human motion sequences from free-form natural language descriptions. This task underpins diverse practical applications, such as humanoid robot control, video game character animation, and virtual reality interactions, driv-

ing increasing research attention in text-to-motion model development in recent years (Sahili et al., 2025).

Evaluating generated motions is essential for advancing T2M research, as reliable assessment supports model improvement across all generative tasks. Traditional metrics (e.g., FID, L1 distance) compare generated motions with reference motions (Guo et al., 2022). However, a single text prompt can validly map to diverse motion sequences, making direct comparison to a single ground-truth reference ineffective. To bypass this, recent "reference-free" evaluators (Voas et al., 2023; Wang et al., 2024) have been trained on text-motion pairs. Yet, these models fall into a data-scarcity trap: the high cost of collecting high-quality 3D motion data hinders their generalization and leads to "hacked" scores that fail to reflect true semantic alignment.

In this work, we propose VeMo, a novel evaluation framework that decouples T2M assessment from the scarcity of 3D motion data. Our core insight is to transform the 3D evaluation problem into a zero-shot video reasoning task: we render generated motions into skinned 2D videos and leverage the vast, internet-scale knowledge of Video-Language Models (VLMs) for assessment. Making use of VLMs, VeMo inherits advanced semantic reasoning capabilities without requiring any motion-specific training. To address the problem of information loss caused by issues such as severe human body self-occlusion when 3D motion is rendered into 2D video, we introduce an entropy-based uncertainty analysis. By calculating the predictive entropy of the VLM across multiple views, VeMo can estimate the reliability of its own scores and select the most informative perspectives for evaluation. This ensures that the 2D proxy remains a faithful representation of the underlying 3D semantics.

To validate T2M evaluators, we contribute a test-only and human-annotated meta-evaluation benchmark. We take the prompts from the HumanML3D (Guo et al., 2022) test set, annotating 2202 diverse text-motion pairs generated from two widely adopted T2M models (Tevet et al., 2022; Jiang et al., 2023), balancing the ratio of matching and non-matching pairs. The benchmark features: 1) coarse-grained alignment labels (denoting overall text-motion match) and 2) fine-grained judgment labels (e.g., Faithfulness and Naturalness). We also design a pipeline (e.g., regenerate controversial motions) to ensure full consistency in oracle annotations.

[1]Zhejiang University [2]Westlake University. Correspondence to: Yue Zhang <zhangyue@westlake.edu.cn>.

*Proceedings of the 43rd International Conference on Machine Learning*, Seoul, South Korea. PMLR 306, 2026. Copyright 2026 by the author(s).

The average of Inter-Annotator Agreement (Krippendorff's Alpha) between oracle annotations and untrained users' annotations exceeds 0.67, demonstrating high data quality.

We compared VeMo with classic reference-based (Tevet et al., 2022) and recent reference-free evaluation methods (Voas et al., 2023; Wang et al., 2024). On our benchmark, VeMo shows the best correlation with human judgments. This confirms that VeMo can provide reliable text-motion alignment assessment without using motion data and human labels, thus addressing existing limitations. Refer to Fig. 1 for a qualitative sense. We summarize our contributions as:

- We use VLMs to evaluate alignment between text and generated motion, enabling internet-scale data to benefit T2M evaluation without the need for T2M data.

- We introduce a meta-evaluation benchmark to assess existing and proposed metrics, validated by user study.

- We show that VeMo outperforms existing metrics in evaluating semantic alignment in T2M generation.

## 2. Related Work

**Text-to-motion** (T2M) aims to synthesize realistic human motion sequences from natural language descriptions. Recent advancements have largely followed two paradigms: discrete-token methods and continuous diffusion models. Discrete approaches (Zhang et al., 2023a; Lou et al., 2023; Jiang et al., 2023; Guo et al., 2024; Chen et al., 2024; Li et al., 2025) use VQ-VAEs (Van Den Oord et al., 2017) to tokenize motion into a "language-like" format, subsequently using Transformers (Vaswani et al., 2017) or LLMs (Brown et al., 2020) for generative modeling. Conversely, continuous latent-space diffusion models bypass quantization to learn complex motion dynamics directly (Chen et al., 2023; Shafir et al., 2023; Zhang et al., 2023b; Tevet et al., 2022; 2024; Uchida et al., 2025). Emerging methods such as Mo-MADiff (Zhang et al., 2025) and LEAD (Andreou et al., 2025) combine discrete and continuous strategies for finer control. The significant strides of T2M models suggest that the evaluation of generated motion is a timely consideration.

**Text-to-motion evaluation.** Traditional metrics rely on reference motions: **FID** calculates scores for each model by comparing distributions of generated and reference motions, rather than for each text-motion pair, which is excluded from the main baselines and analyzed in Appendix (Tab. 10). **L1 distance** measures the distance between joints of the generated and reference motion. As the inherently one-to-many nature of T2M mapping makes direct comparison to a ground-truth reference expensive and impractical, "reference-free" evaluators have emerged. Earlier efforts Tevet et al. (2022; 2024); Han et al. (2025) measure

the **Multimodal (MM) Distance** between text and motion embeddings. **MoBERT** (Voas et al., 2023) trains an evaluation model using fine-grained text-motion labels rather than coarse-grained alignment labels. However, the scarcity of T2M data limits these methods' generalization. VeMo enables internet-scale text-vision data to benefit T2M evaluation without the need for motion data. **MotionCritic** (Wang et al., 2024) estimates text-independent human perceptual preference over motions. **PP-Motion** (Zhao et al., 2025) instead focuses on motion fidelity from the perspective of physical plausibility and feasibility. These directions are complementary to VeMo: rather than judging motion realism or physical validity alone, VeMo evaluates whether a rendered motion semantically matches the input text.

Recent studies have incorporated VLMs into T2M training (Han et al., 2025; Pappa et al., 2024). However, they do not explicitly examine whether VLMs can reliably understand text-motion alignment, and the resulting models are still evaluated using traditional metrics (e.g., MM Distance).

**Meta-evaluation** benchmarks, dedicated resources for comparing empirical usefulness of different evaluation methods, have become foundational in mature generative tasks such as text-to-text/image (Tu et al., 2024; Stufflebeam, 2011; Son et al., 2024). In the field of T2M, while prior works (Voas et al., 2023; Wang et al., 2024) have introduced human-labeled datasets of generated motions, these datasets were used to train their respective evaluation models, not designed to validate the evaluative generalizability of different evaluation methods. This lays the research gap that our benchmark specifically addresses: We only provide a test set and no training set, to compare the generalizability of T2M evaluators. Table 1 features the most related T2M evaluators and human-labeled datasets of generated motions.

## 3. Dataset for Meta Evaluation

We collect generated motions and texts using existing datasets and T2M models (Sec. 3.1); we design a pipeline to collect human annotations (Sec. 3.2), depicted in Fig. 2(a).

### 3.1. Data Collection and Motion Generation

**Data source.** HumanML3D (Guo et al., 2022) is a recent dataset, textually re-annotating motion capture from the AMASS (Mahmood et al., 2019) and HumanAct12 (Guo et al., 2020) collections. It contains 14,616 motions annotated by 44,970 textual descriptions, widely adopted to train T2M models. We take the textual descriptions from the HumanML3D's test set to generate and evaluate motions. To ensure the diversity and representativeness of the evaluation set, we applied a Sentence Transformer (Song et al., 2020) to encode prompts and remove duplicates via hierarchical clustering with a threshold of 0.8. We further

*Table 1.* Landscape of T2M evaluators and generated data with human label. We distinguish VeMo from related works by its ability to evaluate motions zero-shot (Trained on M: ✗) across any motion representation that can be rendered into a video. Our accompanying dataset is the first dedicated strictly to meta-evaluation (✓ Test only), providing higher granularity than previous human-labeled resources.

| | Evaluation Model | | | Generated M w/ Human Label | | |
|---|---|---|---|---|---|---|
| | Input | M-format | Trained on M | Input | Test only | Label granularity |
| Multimodal Distance (Guo et al., 2022) | T,M | Fixed | ✓ | - | - | - |
| MoBERT (Voas et al., 2023) | T,M | Fixed | ✓ | T,M | ✗ | Fine |
| MotionCritic (Wang et al., 2024) | M | Fixed | ✓ | M | ✗ | Coarse |
| VeMo (Ours) | T,M | Any | ✗ | T,M | ✓ | Fine → Coarse |

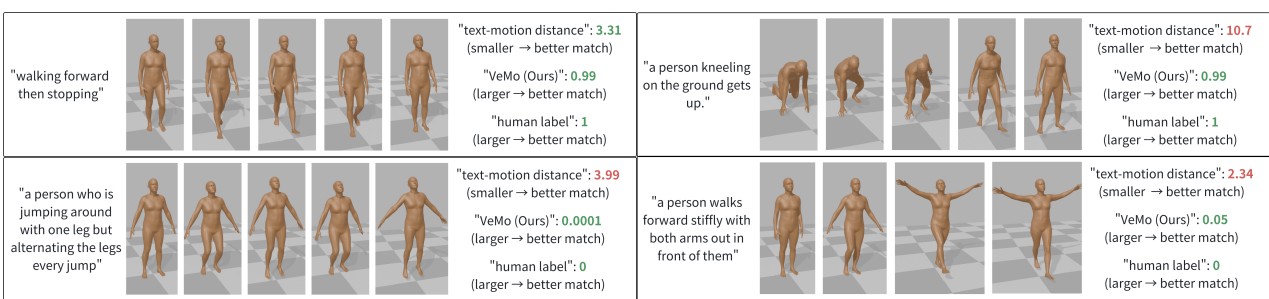

*Figure 1.* Each block displays the text prompt, sampled motion frames, and the resulting evaluation scores. While standard text-motion distance often fails to reflect semantic alignment (in red), our proposed VeMo consistently aligns with human labels (in green).

filter the resulting set by removing prompts with spelling errors or those describing actions outside the scope of the HumanML3D joint set (e.g., dexterous hand movements or gaze). This process yields a final set of 1,101 unique text prompts. Refer to **Appendix** for deduplication details.

**Motion generation.** A T2M model takes textual motion annotations as input and outputs motion sequence $M = (m_t)_{t=1}^{N}$ of human poses represented by joint rotations or positions $m_t \in \mathcal{R}^{J \times D}$. $J$ is the number of joints and $D$ is the dimension of the joint representation. Specifically, we employ a diffusion-based MDM (Tevet et al., 2022) and an autoregressive MotionGPT (Jiang et al., 2023) to generate motion data from 1101 selected prompts for subsequent meta-evaluation. We use these two models because their codebases are widely adopted as the foundation for other methods (Tevet et al., 2024; Han et al., 2025), and both are trained on the HumanML3D's trainset and support the animation of body motions for the 22-joint SMPL human model. Finally, we obtain generated motions from MDM and MotionGPT, resulting in 2,202 text-motion pairs.

**Objective.** After obtaining each pair of text $T$ and generated motion $M$, an evaluation system $\phi$ needs to take $T$ and $M$ as inputs and convert them into a scalar $\phi(T, M)$, which reflects the degree of alignment between $T$ and $M$. The ideal $\phi(T, M)$ is expected to correlate with human annotation.

### 3.2. Motion Visualization and Human Annotation

**Visualization.** To facilitate human assessment, we convert 3D motion data into 2D skinned human videos using

Blender[1]. We optimized the rendering environment and camera movement to ensure that the human model's entire body is fully visible in the video, with clear movements, and that the human model occupies more than 1/10 of the frame. More details are provided in the **Appendices**.

**Annotation pipeline.** To ensure annotation quality, we use a single-choice question format, as forced selection is generally easier and more reliable than direct rating (Kendall, 1948; Wang et al., 2024). We evaluate each text-video pair using two fine-grained criteria: **T2M Faithfulness**, which asks whether the human in the video executes the text-described action completely, accurately, and in the correct order, and **Motion Naturalness**, which asks whether the motion is natural and free from joint distortion or unnatural movements beyond the text description. Both criteria are answered with "Yes" or "No". To ensure high data quality, oracle annotators discuss and resolve inconsistent cases, or regenerate motions for re-annotation when necessary, until full consensus is reached. Approximately 12% of samples triggered regeneration, typically because normal motion patterns were interleaved with abnormal or ambiguous movements. We further validate the oracle annotations through independent user studies in Section 5.4.

**Label aggregation.** We aggregate these fine-grained judgments into a coarse-grained Alignment label. A pair is labeled as aligned (positive) only if it satisfies both Faithfulness and Naturalness; otherwise, it is labeled as unaligned

---

[1]Blender is a popular open-source 3D creation suite, refer to https://www.blender.org/ for details

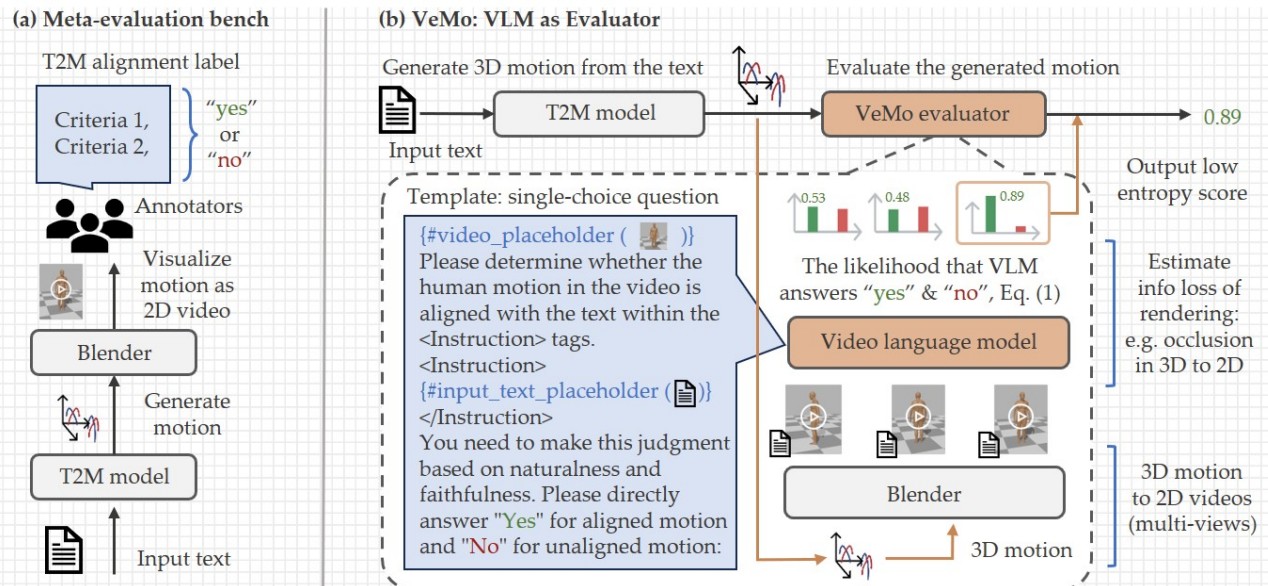

*Figure 2.* (a) Meta-evaluation benchmark: annotators label whether the generated motion matches the corresponding text or not. (b) VeMo framework: a zero-shot evaluator leveraging multi-view rendering, VLMs, and entropy-based uncertainty estimation for reliable scoring.

*Table 2.* Number of positive/negative labels on two data splits.

|          | Alignment | Faithfulness | Naturalness |
|----------|-----------|--------------|-------------|
| MDM      | 613/1101  | 621/1101     | 1069/1101   |
| MotionGPT| 506/1101  | 528/1101     | 1046/1101   |

(negative). As shown in Table 2, this yields a balanced dataset for evaluating the generalizability of automatic metrics, since unnatural samples are scarce. Note that the prior evaluation model (Voas et al., 2023) is developed using human labels for Faithfulness and Naturalness. Our datasets can therefore serve as a test set for studying its empirical usefulness, i.e., generalizability. Although our annotation pipeline supports additional criteria, this work focuses on the above two, which we consider most important.

## 4. Video Language Model as Evaluation Model

We first formulate how to convert paired text and motion into normalized VLM scores (Sec. 4.1). Then, we devise an *entropy-based technique to ensure high-quality VLM scores* with low information loss when 3D motion is rendered into video (Sec. 4.2). The process is shown in Fig. 2(b).

### 4.1. Formulate VLM Scores

A text-to-motion model T2M(·) takes a textual prompt $T$ as the input and outputs a motion sequence $M = (m_t)_{t=1}^{N}$, where $m_t$ is a pose vector at timestep $t$, encoding joint angles, positions, etc. We denote $I$ as the instruction template (Fig. 2 (b)) and denote random variable $Y$ as the candidate answer, taking values from $\{y^+ = \text{``yes''}, y^- = \text{``no''}\}$.

To compute the alignment score with VLM, we first use Blender software to render the motion $M$ into a video $V = (v_t)_{t=1}^{N}$, using the same environment as in Section 3.2. A pretrained VLM (e.g., InternVL3-14B (Zhu et al., 2025)) then outputs the likelihood of $y \in Y$ with $I, T$ and $V$. Finally, we aggregate the likelihood into a normalized distribution representing alignment score:

$$\mathcal{P}_{\text{VLM}}(y^+|I,T,V) = \frac{\text{LH}(y^+|I,T,V)}{\text{LH}(y^-|I,T,V) + \text{LH}(y^+|I,T,V)} \tag{1}$$

where LH is conditional likelihood, output by VLM. As exemplified in Figure 2 (b), we take *"yes"* as $y^+$, which refers to alignment between text and motion; we take *"no"* as $y^-$, which refers to misalignment between text and motion. Notably, evaluating T2M models with VLMs involves rendering 3D information to 2D information, where accumulated biases and noise (e.g., single-view occlusion) may hamper the quality of VLM scores. To this end, we do not sample the hard prediction (i.e., words) from VLM's continuous output (i.e., likelihood). The likelihood reflects the VLM's confidence in the answer and can help us estimate the information loss, we will discuss later.

### 4.2. Select Low-Entropy Scores for Evaluation

Large language models suffer from the notorious hallucination problem (Huang et al., 2025), and the same is true for VLMs (Liu et al., 2024) — the model may also fabricate answers, even if a definite judgment cannot be drawn from the data input to the model. Fortunately, recent research (Farquhar et al., 2024) has revealed that there is a strong

correlation between hallucination and the entropy of the model's output, with speculative hallucinations typically occurring alongside high entropy.

Intuitively, when a rendered video loses important 3D information, such as when the lower body is occluded, we can only guess whether the person in the video is performing a specific leg movement, which is a case of speculative hallucinations. Inspired by the success in speculative hallucination detection, we estimate whether an input rendered video contains sufficient information to answer the textual question by calculating the entropy as follows:

$$H(Y|I, T, V) = \\ -\sum_{y \in Y} \mathcal{P}_{\text{VLM}}(y|I, T, V) \log\left[\mathcal{P}_{\text{VLM}}(y|I, T, V)\right] \quad (2)$$

Based on Eq. (2), we render each motion $M$ into $K$ videos $(V^i)_{i=1}^K$ from different views, and take the final evaluation score $S_{\text{VLM}}$ for each text-motion pair as follows:

$$S_{\text{VLM}}(T, M) = \mathcal{P}_{\text{VLM}}(y^+|I, T, V'), \\ V' = \underset{V \in (V^i)_{i=1}^K}{\arg\min} H(Y|I, T, V) \quad (3)$$

### 4.3. Theoretical Analysis

Optimal view selection maximizes the mutual information $I(Y; V|T)$ between the alignment judgment $Y$ and the rendered view $V$. This is equivalent to minimizing conditional entropy $H(Y|V, T)$, since $I(Y; V|T) = H(Y|T) - H(Y|V, T)$ and $H(Y|T)$ is constant. Figure 3 illustrates the non-monotonic relationship between signal degradation and predictive entropy. In the Entropy Increase Phase ($0.0 \leq$ noise $< 0.6$), increasing occlusion introduces aleatoric uncertainty, as the VLM must speculate on hidden 3D joint movements, leading to a rise in entropy. Conversely, in the Rejection Phase (noise $\geq 0.6$), extreme degradation triggers "confident rejection": the human subject becomes indistinguishable from background noise, causing the model to assign a near-certain probability to the "No" label (misalignment), which paradoxically lowers the entropy. This explains our design: we make the human body appear in the rendered video, to ensure the video is informative enough for the VLM to work in the entropy-increase phase. We also validate the entropy-based view selection in Sec. 5.4.

## 5. Experiments

We first detail the experimental settings (Sec. 5.1) and baseline metrics for comparison (Sec. 5.2). Then, we compare VeMo with existing automatic metrics on our meta-evaluation benchmark (Sec. 5.3). Finally, we validate our key designs and provide deeper analysis in Sec. 5.4.

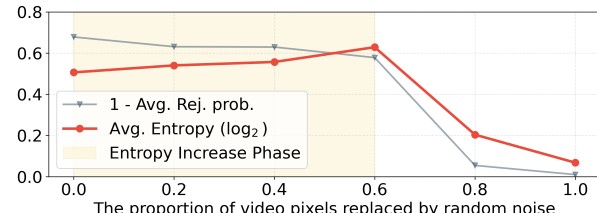

*Figure 3.* Impact of simulated occlusion on VLM predictive entropy. Pixels are randomly replaced with Gaussian noise.

### 5.1. Experimental Settings

**Implementation details.** We conducted experiments on 1×A100-80G GPU, using LabelStudio (Tkachenko et al., 2020-2025) as frontend for user study, detailed in Appendix. We converted the **Alignment** labels derived from human judgments (see Section 3.2) into binary labels for subsequent meta-evaluation: 0 indicates that the generated motion is not aligned with the prompt used to generate it, while 1 indicates that the generated motion is aligned with its prompt. The meta-evaluation dataset (Sec. 3) is only used for testing, unlike previous works Voas et al. (2023); Wang et al. (2024) where it was also used to fine-tune evaluation models. We take InternVL3-14B (Zhu et al., 2025) as our base in main experiments, and use VeMo as a zero-shot evaluation model, without the use of any text-motion pairs for training or any human label for in-context learning (Dong et al., 2022).

**Datasets and metrics.** We conducted meta-evaluation experiments on the dataset detailed in Section 3, aiming to identify which evaluation scores are most suitable for evaluating text-motion alignment. To this end, we measure the correlation between different evaluation scores and human judgment using the following metrics: 1) **AUC-ROC** measures a binary classifier's ability to distinguish positive from negative classes, defined as the area under the ROC curve (Metz, 1978). ROC curve plots True Positive Rate (TPR) against False Positive Rate (FPR) across all possible decision thresholds. 2) **AUPR** assesses classifier performance (especially informative for imbalanced data) as the area under the Precision-Recall curve (plots Precision vs. Recall). 3) **Kendall's $\tau$** (Kendall, 1945; 1948) measures the correspondence between evaluation scores-based ranking and human scores-based ranking. Values close to 1/-1 indicate strong agreement/disagreement. 4) **Spearman's $\rho$** (Zwillinger & Kokoska, 1999) measures the monotonic association between evaluation scores-based ranking and human scores-based ranking. This statistic varies between -1 and +1 with 0 implying no correlation. 5) **KS** (Kolmogorov–Smirnov) test statistic (Massey Jr, 1951) quantifies the maximum separation between the cumulative distribution functions of evaluation scores with positive/negative human labels. 6) **Mann-Whitney U Test (p-value)** (McKnight & Najab,

*Table 3.* Results of the correlation between different evaluation scores and human judgements, reported on meta-evaluation dataset. ↑ means larger is better, ↓ means lower is better.

| Evaluation Method | AUC-ROC ↑ | AUPR ↑ | KS ↑ | $\tau$ ↑ | $\rho$ ↑ | p-value ↓ |
|---|---|---|---|---|---|---|
| *(reference-based automatic evaluation method)* | | | | | | |
| Minus L1 Distance | 0.627 | 0.628 | 0.204 | 0.180 | 0.220 | <1e-6 |
| *(reference-free automatic evaluation method)* | | | | | | |
| Minus Multimodal Distance | 0.513 | 0.521 | 0.048 | 0.018 | 0.022 | 0.149 |
| R@1-Precision | 0.508 | 0.559 | 0.016 | 0.024 | 0.024 | 0.129 |
| R@2-Precision | 0.504 | 0.573 | 0.007 | 0.009 | 0.009 | 0.343 |
| R@3-Precision | 0.498 | 0.590 | 0.005 | -0.005 | -0.005 | 0.595 |
| MoBERT-base | 0.526 | 0.541 | 0.057 | 0.037 | 0.045 | 0.018 |
| MoBERT-F | 0.532 | 0.548 | 0.087 | 0.045 | 0.055 | 0.005 |
| MoBERT-N | 0.549 | 0.551 | 0.088 | 0.069 | 0.084 | 4e-05 |
| MoBERT-max(F/N) | 0.549 | 0.553 | 0.088 | 0.069 | 0.084 | 4e-05 |
| MoBERT-min(F/N) | 0.534 | 0.549 | 0.086 | 0.048 | 0.058 | 0.003 |
| MotionCritic | 0.506 | 0.517 | 0.027 | 0.009 | 0.010 | 0.312 |
| **VeMo (Ours)** | **0.720** | **0.743** | **0.354** | **0.311** | **0.381** | **<1e-6** |
| User-1 Score | 0.829 | 0.878 | 0.658 | 0.659 | 0.659 | <1e-6 |
| User-2 Score | 0.835 | 0.876 | 0.670 | 0.677 | 0.677 | <1e-6 |

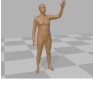
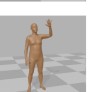

T: "a person waves a friendly hello."
"GT human label": **1**
"VeMo (Ours)": **0.999**
"MM Distance": 11.135
"MoBERT": 0.373
"MotionCritic": -11.257

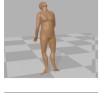
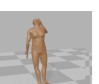

T: "a person opens and drinks from a container."
"GT human label": **0**
"VeMo (Ours)": **0.013**
"MM Distance": 8.609
"MoBERT": 0.418
"MotionCritic": -7.034

*Figure 4.* Case study. Evaluation scores from VeMo and baseline methods on test prompts compared against ground truth human labels.

2010) is a nonparametric test. Its null hypothesis is that the positive and negative human-labeled score distributions are identical, and the alternative hypothesis is that the positive human-labeled score distribution is greater.

### 5.2. Baselines.

We compare VeMo against the established T2M metrics. For distance-based metrics, we report the negative values to ensure that higher scores consistently denote superior T2M alignment, matching the direction of human judgments.

**L1 Distance** measures the average physical displacement between joints of the generated motion and a single ground-truth reference. While precise, its utility is constrained by the scarcity of comprehensive 3D reference data.

**Multimodal Distance (MM Dist)** calculates the Euclidean distance between text and motion embeddings in a shared feature space. We use the widely adopted biencoder (Tevet et al., 2022) to encode motion and text.

**R@K-Precision** formulates evaluation as a retrieval task. It measures the percentage of instances where the ground-truth text is among the top-$K$ (1/2/3) closest matches to a generated motion within a pool of 31 mismatched texts.

**MoBERT** (Voas et al., 2023) is a specialized evaluation model trained on T2M data. We include their open-source evaluator: MoBERT-base (an evaluation model trained on real text-motion pairs), as well as versions fine-tuned on generated text-motion pairs with Faithfulness (MoBERT-F) and Naturalness (MoBERT-N) human labels.

**MotionCritic** (Wang et al., 2024) is a text-independent motion evaluator, trained to align with human-preferred rankings of motions. The evaluation scores indicate whether the motion is more in line with human preferences rather

than the T2M distance. We take their open-source evaluator as a baseline without re-implementation, for reference only.

### 5.3. Main Results

The empirical evaluation of VeMo against existing automatic metrics is detailed in Table 3, adhering to their official implementations. We **bold** and underline the best and the second-best results. VeMo consistently achieves state-of-the-art performance across all six meta-evaluation metrics, demonstrating a superior correlation with human judgment despite its zero-shot nature and lack of exposure to motion-specific training data. Among reference-free methods, the maximum improvement of VeMo in the KS statistic is more than 4 times that of the best alternative, and the p-value $< 1e-6$, indicating that the scores with positive human labels are generally and significantly higher than those with negative human labels; In terms of the correlation coefficients $\tau$ and $\rho$, VeMo also achieves a 4-fold improvement, highlighting a strong correlation between VeMo and human judgments; AUC-ROC and AUPR measure performance of evaluation scores used in binary classification across a wider range of thresholds, and VeMo also achieves top-1 performance, with an improvement of up to 0.171. Based on the results, we also make a few comparisons as follows.

First, a lower KS statistic (0.048) indicates that the distribution difference between the **Multimodal Distance** with positive human labels and that with negative human labels is small. The $\tau$ and $\rho$ coefficients are close to 0, indicating that the Minus Multimodal Distance has a very weak correlation with human labels, and using Multimodal Distance as an evaluation score to assess whether the text and generated motion sequence are aligned is not trustworthy. The standard **R@K-Precision** is calculated based on the Minus Multimodal Distance, so we arrive at the same conclusion.

*Table 4.* Ablation studies on number of input frames and VLMs used in VeMo.

| Backbone VLM used in VeMo | view selection strategy | number of frames | AUC-ROC ↑ | AUPR ↑ | KS ↑ | $\tau$ ↑ | $\rho$ ↑ | p-value ↓ |
|---|---|---|---|---|---|---|---|---|
| *(Impact of view selection)* | | | | | | | | |
| InternVL3-14B | human-opt view | 32 | **0.723** | 0.740 | 0.342 | **0.315** | **0.385** | <1e-6 |
| InternVL3-14B | min entropy view | 32 | 0.720 | **0.743** | **0.354** | 0.311 | 0.381 | <1e-6 |
| InternVL3-14B | random view | 32 | 0.711 | 0.734 | 0.322 | 0.299 | 0.366 | <1e-6 |
| InternVL3-14B | max entropy view | 32 | 0.706 | 0.722 | 0.319 | 0.291 | 0.356 | <1e-6 |
| *(Multimodal Models Supporting Video Input)* | | | | | | | | |
| InternVL3-14B | human-opt view | 32 | 0.723 | 0.740 | 0.342 | 0.315 | 0.385 | <1e-6 |
| InternVL3-14B | human-opt view | 8 | 0.720 | 0.738 | 0.343 | 0.311 | 0.381 | <1e-6 |
| InternVL3.5-14B | human-opt view | 32 | 0.709 | 0.734 | 0.331 | 0.296 | 0.363 | <1e-6 |
| InternVL3.5-14B | human-opt view | 8 | 0.698 | 0.721 | 0.306 | 0.281 | 0.343 | <1e-6 |
| InternVL3-8B | human-opt view | 32 | 0.687 | 0.706 | 0.291 | 0.264 | 0.324 | <1e-6 |
| InternVL3-8B | human-opt view | 8 | 0.683 | 0.701 | 0.284 | 0.258 | 0.316 | <1e-6 |
| *(Video-Text Foundation Models)* | | | | | | | | |
| InternVideo2.5 | human-opt view | 128 | 0.684 | 0.700 | 0.292 | 0.260 | 0.318 | <1e-6 |
| InternVideo2.5 | human-opt view | 32 | 0.688 | 0.706 | 0.292 | 0.266 | 0.325 | <1e-6 |
| InternVideo2.5 | human-opt view | 8 | 0.669 | 0.688 | 0.263 | 0.239 | 0.292 | <1e-6 |
| *(Video-Text Representation Learning Model)* | | | | | | | | |
| ViCLIP-L-14 | human-opt view | 8 | 0.559 | 0.543 | 0.099 | 0.084 | 0.103 | 1e-06 |

*Table 5.* User study: agreement between individual users and oracle annotators, measured by Krippendorff's $\alpha$ coefficients.

| | Alignment | Faithfulness | Naturalness |
|---|---|---|---|
| User-1 | 0.6566 | 0.6376 | 0.7356 |
| User-2 | 0.6681 | 0.6564 | 0.7896 |

Second, **MoBERT** is the second-performing reference-free method. MoBERT-base initially exhibits a negative correlation with human judgments and human-feedback tuning (MoBERT-F/N) improves its performance. However, compared to VeMo, which neither incorporates any human feedback on motion nor has been trained on motions, MoBERT still has lower discriminative power in distinguishing positive and negative human labels. We also notice that Motion-Critic, an evaluation model that only takes generated motion as input, performs worse than the classic multimodal distance, indicating that it is necessary to consider both the text and the generated motion when evaluating T2M alignment.

Third, the $\tau$ and $\rho$ coefficients for **users' scores** are close to 0.7, outperforming all other evaluation scores, indicating that no automatic metric can yet fully replace human evaluations. Furthermore, the coefficients of reference-free baselines are less than 0.1, and the $\rho$ coefficient of reference-based **L1 distance** exceeds 0.2, indicating that existing reference-free methods not only have a weak correlation with human evaluations but also cannot replace reference-based methods. In contrast, $\tau$ and $\rho$ for VeMo exceed 0.3, demonstrating that our approach not only achieves a meaningful correlation with human evaluations but also can outperform reference-based evaluation methods.

### 5.4. Analysis

**Inter-annotator agreements (IAA)** between labels from independent user studies and labels from oracle annotators

are measured by Krippendorff's Alpha (Krippendorff, 2011). Users independently annotated the text-video data of our benchmark in accordance with the specifications in Section 3.2 without discussing with each other. As shown in Table 5, we achieve high inter-annotator agreement. This conclusion is consistent with the finding regarding the performance of user scores from the main experiment presented in Table 3. LabelStudio (Tkachenko et al., 2020-2025) is used as the frontend for user annotation, see **Appendix** for more details.

**Entropy-based view selection** (Sec. 4.2) is validated in Table 4. We use the human-optimized rendering view (Sec. 3.2) as $V^0$ (*human-opt view*) and obtain another view $V^1$ by randomly rotating the camera around the human body. We then evaluate four view-selection strategies: randomly choosing one view from $\{V^0, V^1\}$ (*random view*), selecting the view with the highest predictive entropy (*max entropy view*), selecting the view with the lowest predictive entropy (*min entropy view*), and directly using $V^0$. The min-entropy view performs close to the human-opt view and consistently outperforms the random and max-entropy views. This suggests that entropy-based selection provides a practical automatic alternative when a carefully engineered rendering protocol is unavailable or when the evaluator is transferred to new rendering settings. We therefore treat low-entropy selection not as a replacement for the human-opt view, but as an uncertainty-aware approximation that reduces the manual cost of configuring rendering views and provides a useful reliability signal for VLM-based evaluation. Detailed rendering settings are provided in the **Appendix**.

**Impact of VLMs used in VeMo.** In Table 4, InternVL3-14B consistently achieves top-1 performance across all metrics. Because the APIs for private VLMs are expensive, difficult to reproduce, and typically inaccessible in terms of per-token likelihood, we only consider the following

*Table 6.* Extending VeMo score to HumanML3D benchmark for T2M model evaluation.

| T2M Model | VeMo ↑ | FID ↓ | MM Dist ↓ | R@1 ↑ | R@2 ↑ | R@3 ↑ |
|---|---|---|---|---|---|---|
| MotionGPT (Jiang et al., 2023) | $0.5723^{\pm 0.0034}$ | $0.232^{\pm 0.008}$ | $3.096^{\pm 0.008}$ | $0.492^{\pm 0.003}$ | $0.681^{\pm 0.003}$ | $0.778^{\pm 0.002}$ |
| StableMofus. (Huang et al., 2024) | $0.6528^{\pm 0.0001}$ | $0.098^{\pm 0.003}$ | $2.770^{\pm 0.006}$ | $0.553^{\pm 0.003}$ | $0.748^{\pm 0.002}$ | $0.841^{\pm 0.002}$ |
| MLD-M (Dai et al., 2024) | $0.6626^{\pm 0.0002}$ | $0.073^{\pm 0.003}$ | $2.810^{\pm 0.008}$ | $0.548^{\pm 0.003}$ | $0.738^{\pm 0.003}$ | $0.829^{\pm 0.002}$ |
| MotionLCM-V2 (Dai et al., 2024) | $0.6638^{\pm 0.0005}$ | $0.072^{\pm 0.003}$ | $2.767^{\pm 0.007}$ | $0.546^{\pm 0.003}$ | $0.743^{\pm 0.002}$ | $0.837^{\pm 0.002}$ |
| Real | $0.6825^{\pm 0.0000}$ | $0.002^{\pm 0.000}$ | $2.974^{\pm 0.008}$ | $0.511^{\pm 0.003}$ | $0.703^{\pm 0.003}$ | $0.797^{\pm 0.002}$ |

*Table 7.* Performance comparison of minimum-entropy vs. maximum-entropy based view selection methods.

| | Minimum-Entropy Based Selection | | | | | | Maximum-Entropy Based Selection | | | | |
|---|---|---|---|---|---|---|---|---|---|---|---|
| num views | AUC-ROC | AUPR | KS | $\tau$ | $\rho$ | num views | AUC-ROC | AUPR | KS | $\tau$ | $\rho$ |
| $K$=1 | 0.706 | 0.729 | 0.304 | 0.291 | 0.357 | $K$=1 | 0.706 | 0.729 | 0.304 | 0.291 | 0.357 |
| $K$=2 | 0.721 | 0.742 | 0.346 | 0.312 | 0.382 | $K$=2 | 0.711 | 0.728 | 0.324 | 0.298 | 0.365 |
| $K$=3 | 0.719 | 0.742 | 0.349 | 0.310 | 0.380 | $K$=3 | 0.705 | 0.720 | 0.312 | 0.290 | 0.355 |
| $K$=4 | 0.721 | 0.746 | 0.356 | 0.313 | 0.384 | $K$=4 | 0.701 | 0.718 | 0.308 | 0.284 | 0.347 |
| $K$=5 | 0.723 | 0.748 | 0.361 | 0.315 | 0.386 | $K$=5 | 0.699 | 0.715 | 0.308 | 0.282 | 0.345 |
| $K$=6 | 0.721 | 0.745 | 0.358 | 0.312 | 0.382 | $K$=6 | 0.699 | 0.712 | 0.308 | 0.281 | 0.344 |

open-source VLMs: (1) *Multimodal Models Supporting Video Input* trained on extensive multimodal data (e.g., images, text, tool interactions), which can jointly encode text and video; (2) *Video-Text Foundation Models* focusing on video temporal reasoning, which can also jointly encode text and video; (3) *Video-Text Representation Learning Models*, which encode text and video into vectors independently and finally compute the inner product score. The results show that Video-Text Representation Learning Models exhibit significantly poor performance, with both $\tau$ and $\rho$ coefficients being far below 0.3; In contrast, the coefficients of both *Multimodal Models Supporting Video Input* and *Video-Text Foundation Models* are around 0.3, which demonstrates the importance of jointly encoding text and video.

**Impact of VLMs' input frame number** on VeMo performance. Table 4 shows the results for VeMo using different numbers of frames uniformly sampled from input video. In our experimental setup, exceeding 32 frames results in Out of Memory for InternVL3-14B/8B (Zhu et al., 2025) and InternVL3.5-14B (Wang et al., 2025a); the InternVideo2.5 (Wang et al., 2025b) model has only 8B parameters and supports an input of 128 frames, while ViCLIP-L-14 (Wang et al., 2023) only supports an input of 8 frames. Overall, using more frames leads to a slight improvement in performance. However, when 128 frames are input, Intern-Video2.5 exhibits a slight performance drop, indicating the presence of saturation. Nevertheless, the magnitude of these performance changes is relatively small and does not affect the conclusions drawn from the main experiment.

**Benchmark of T2M approaches.** We base VeMo on InternVL3-14B (32-frame) with human-opt view selection and report VeMo scores on the HumanML3D benchmark covering representative T2M models to give the community a clear baseline for comparison. As shown in Table 6, we observed that several T2M models outperform the ground truth on certain reference-free metrics (e.g., MM Dist and R@K-Precision), which suggests those evaluators can be overfit or "hacked." FID is a reference-based method and has a high correlation with the VeMo metric.

**Impact of K.** We pre-extract six views per motion by uniformly rotating the camera around the body (the engineered "human-opt" view plus five random rotations). For evaluation we sample K views without replacement from these six, and report metrics for min-entropy and max-entropy (Table 7) based view selections. These results lead to three observations. (1) Selecting the min-entropy view (i.e., the most confident view) consistently outperforms selecting the max-entropy view when $K > 1$. (2) The largest gain occurs when moving from $K = 1$ to $K = 2$; beyond $K = 2$ performance quickly saturates. Thus $K = 2$ provides a strong balance between reliability and cost. (3) Generating additional views incurs roughly 30 seconds of rendering per extra perspective per motion. Given this cost, we recommend using the engineered human-opt view (i.e., human-opt, $K = 1$), which already achieves performance close to a higher-$K$ regime.

**Case studies.** Fig. 4 shows two cases where the evaluation scores from VeMo clearly align with human labels, while those from other methods are ambiguous. The prompt (on the top) contains abstract concepts, requiring an understanding that people usually wave their hands to express a friendly "hello". VeMo **faithfully** grasps the underlying action, assigning a high confidence score that aligns with the human label, while the baseline does not. The person in the bottom case drinks water with the back of their head **unnaturally**. VeMo recognizes that "anti-human style" of the human motion, thus assigning a low alignment score. This further demonstrates that VeMo can understand the complex semantics and human style in T2M evaluation. Refer to the

*Table 8.* Evaluation of T2M evaluators on the no-regeneration split containing motions from StableMoFusion and MotionLCM-V2.

| Evaluation Method | AUC-ROC ↑ | AUPR ↑ | KS ↑ | τ ↑ | ρ ↑ |
|---|---|---|---|---|---|
| Baselines | | | | | |
| Minus L1 Dist. | 0.568 | 0.696 | 0.112 | 0.091 | 0.112 |
| Minus MM Distance | 0.519 | 0.658 | 0.046 | 0.026 | 0.032 |
| R@1-Precision | 0.512 | 0.723 | 0.024 | 0.023 | 0.023 |
| R@2-Precision | 0.514 | 0.752 | 0.028 | 0.027 | 0.027 |
| R@3-Precision | 0.516 | 0.773 | 0.032 | 0.032 | 0.032 |
| VeMo | | | | | |
| human-opt view | 0.621 | 0.765 | 0.212 | 0.163 | 0.200 |
| min-entropy view | **0.625** | **0.770** | **0.226** | **0.169** | **0.207** |

*Table 9.* Prompt robustness analysis. The paraphrased prompts preserve the original meaning but alter surface wording.

| Setting | AUC-ROC ↑ | AUPR ↑ | KS ↑ | τ ↑ | ρ ↑ |
|---|---|---|---|---|---|
| default prompt | 0.723 | 0.740 | 0.342 | 0.315 | 0.385 |
| paraphrased prompt | 0.708 | 0.731 | 0.322 | 0.294 | 0.360 |

project repository for more qualitative analysis.

**Expanded the meta-evaluation without regeneration.** We further evaluate all metrics on the newly added StableMo-Fusion (Huang et al., 2024) and MotionLCM-V2 (Dai et al., 2024) split, where generated motions are annotated without regeneration. This split is more challenging because ambiguous outputs are preserved. As in Table 8, the performance of several baselines decreases on this split, while VeMo maintains the strongest correlation with human judgments. These results suggest that the VLM-based evaluation remains effective beyond the original motion generators.

**Prompt robustness.** Prompting is not a key design component of the VeMo pipeline, and we do not perform prompt optimization for any VLM. We first draft the instruction template for human meta-evaluation and then apply the same template directly to all VLMs. The results across a wide range of VLMs in Tab. 4 succinctly reflect the robustness. We intentionally avoid over-engineering the prompt, since a stronger VLM with human-level video-language understanding should be able to handle natural human-centric instructions. To further examine sensitivity to textual phrasing, we paraphrase the input prompts using GPT-4o-mini (Hurst et al., 2024) before feeding them into the VLM. As shown in Table 9, paraphrasing leads to a slight performance decrease, but VeMo still substantially outperforms the best reference-free baseline. This indicates that VeMo's performance mainly comes from the VLM's video-language reasoning ability rather than a highly optimized prompt.

**Rendering sensitivity and efficiency.** Since VeMo evaluates a 2D visual proxy of 3D motion, rendering quality affects both accuracy and cost. Replacing the default skinned rendering with stick-figure rendering (A.Tab. 13) reduces AUC from 0.723 to 0.608 and Spearman's $\rho$ from 0.385 to 0.187, indicating that body appearance and visual clarity help VLM-based evaluation. Still, the stick-figure variant outperforms the best reference-free baseline, suggesting that lightweight visualizations preserve useful temporal joint cues. We also find that rendering, rather than VLM inference, is the main time bottleneck: Joint-to-Mesh conversion and Mesh-to-Video rendering take 182s and 31s per motion,

respectively, while InternVL3-14B inference with 32 frames takes 1.597s per video. For efficient evaluation, stick-figure visualization skips Joint-to-Mesh conversion and takes only 0.007s per frame; combined with InternVL3-1B, it processes a 100-frame motion in under one second. These results show a practical speed-accuracy trade-off: full-body rendering is more accurate, while stick-figure rendering offers a fast alternative for large-scale screening.

**More in-depth analysis** of VeMo is provided in the **Appendix**. This covers detailed efficiency trade-offs, acceleration strategies, the stability of VLM scores, limitations of entropy-based view selection, multi-view integration, etc. Together, these analyses clarify when VeMo is reliable, where rendering or VLM perception may introduce failure cases, and how the proposed framework can be extended.

## 6. Conclusion

We considered the evaluation of T2M alignment, proposed a new meta-evaluation benchmark to solve the problem that there is no shared testbed to fairly compare the generalizability of T2M evaluators. We use the VLM to solve T2M evaluation, and devise an entropy-based technique to foster a high-quality VLM score when 3D motion is rendered into 2D video. Our method, VeMo, brings internet-scale text-vision knowledge to T2M evaluation and achieves evaluation performance closer to human level. This evaluation method can potentially not only provide a fairer comparison for different T2M models but also offer more accurate feedback for the development of new models. *Additional results, discussions, and limitations can be found in the Appendix.*

## Acknowledgements

This work has emanated from research conducted with the financial support of the National Natural Science Foundation of China Key Program under Grant Number 62336006.

## Impact Statement

This paper presents work whose goal is to advance the field of machine learning. There are many potential societal consequences of our work, none of which we feel must be specifically highlighted here beyond standard evaluation.

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

# A. Appendix

## A.1. Details on Benchmark Data Collection

**Data source.** We based our benchmark data on HumanML3D (Guo et al., 2022), a recent dataset. HumanML3D textually re-annotates motion capture from the AMASS (Mahmood et al., 2019) and HumanAct12 (Guo et al., 2020) collections. It contains 14,616 motions annotated by 44,970 textual descriptions, split in train, val, test sets. The train and val splits of HumanML3D are widely adopted to train T2M models. We take the descriptions from the HumanML3D's test set as prompts to generate and evaluate motions. To ensure the diversity and representativeness of the prompts, we use Sentence Transformer, i.e., all-mpnet-base-v2 (Song et al., 2020), to encode all prompts for deduplication. We recursively merged pairs of clusters of sample data, using the cosine distance given by these embeddings, with a clustering threshold of 0.8, resulting in approximately 1.5k prompts. Finally, we customized more conditions to further filter the remaining 1.5k prompts. The removal conditions include spelling errors in the action descriptors in the prompts, or prompts describing dexterous hand movements, gaze, and other actions that do not belong to the HumanML3D joints. In the end, 1101 texts remained.

**Motion generation.** A trained T2M model will take textual motion annotations as input and output motion sequence $M = (m_t)_{t=1}^{N}$ of human poses represented by joint rotations or positions $m_t \in \mathcal{R}^{J \times D}$. $J$ is the number of joints and $D$ is the dimension of the joint representation. Specifically, we employ a diffusion-based MDM (Tevet et al., 2022) and an autoregressive MotionGPT (Jiang et al., 2023) to generate motion data from 1101 selected prompts for subsequent meta-evaluation. Because the codebases of these two models are widely adopted as the foundation for other methods (Tevet et al., 2024; Han et al., 2025), and both models are trained on the HumanML3D's trainset and support the animation of body actions for the 22-joint SMPL human model. For MDM, we use the official checkpoint "humanml_trans_dec_512_bert-50steps"; for MotionGPT, we also use the official checkpoint "OpenMotionLab/MotionGPT-base". Finally, we obtain generated motions from MDM and MotionGPT, and there are a total of 2202 text-motion pairs.

## A.2. Details on Visualization Settings

To annotate a generated motion sequence $M$ and the text used for generation, we first use Blender software to convert the generated motion $M$ into skinned human model video $V$. We first use the smplx Python package to create a neutral SMPL model (Loper et al., 2023). Then, we run SMPLify (Bogo et al., 2016) to convert the motion sequence into a 3D voxel (.obj file), which can be directly imported into the Blender environment and rendered into a video. We configure the rendering resolution to 1088×1088 pixels, with PNG set as the output format for intermediate frame images to ensure high-quality image data for subsequent video compilation. The scene background is configured as a natural white color (RGB: 1.0, 1.0, 1.0) with an intensity value of 0.6, which avoids overexposure while ensuring the human model stands out clearly against the background. A chessboard-patterned floor is added to the scene to provide spatial reference and enhance visual layering. This floor has a size of 10×10 units and is divided into 10×10 grid divisions; its vertical position (Z-axis) is aligned with the lowest point of the human model's bounding box to ensure it fits naturally under the model. The floor uses a semi-transparent material (transparency set to 0.5) based on the Principled BSDF shader, and a chessboard texture is applied via UV smart projection to ensure the pattern is evenly distributed and displayed correctly. Additionally, the floor is set to be unselectable to prevent accidental modification during the rendering process. For the human model, a custom "DarkBronzeSkinMaterial" is developed to simulate a realistic skin-like appearance. The material's base color is set to an RGB value of (0.3, 0.15, 0.07) (a deep brown with warm bronze undertones), the metallic attribute is adjusted to 0.4 to enhance subtle reflective properties, and the roughness is set to 0.6 to soften excessive gloss, resulting in a natural texture that better showcases the model's contour details and motion changes.

Lighting is provided by a single SUN-type light source with an energy value of 4.5 to ensure sufficient and uniform illumination of the human model. The light source is fixed at the spatial position (-4, -6, 6) and rotated to face the geometric center of the human model—this rotation is calculated by converting the vector from the light source to the model's center into an Euler angle, ensuring the light rays are directed toward the model and minimizing harsh shadows that could obscure motion details. For camera configuration, a new camera is created for each frame of the motion sequence to maintain consistent framing of the human model. The camera's position is determined by offsetting the human model's geometric center by a fixed vector (-1, -3, 0.6); similarly to the light source, the camera is rotated to face the model's center (using the vector from the camera to the model's center converted to an Euler angle). This setup ensures that the human model's entire body remains fully visible in each frame, and the model occupies more than 1/10 of the frame area as required. After rendering all motion frames into individual PNG images, the images are compiled into a video file using the H.264 codec (libx264) with a frame rate of 20 FPS and a quality parameter of 9. This codec and parameter combination balances video

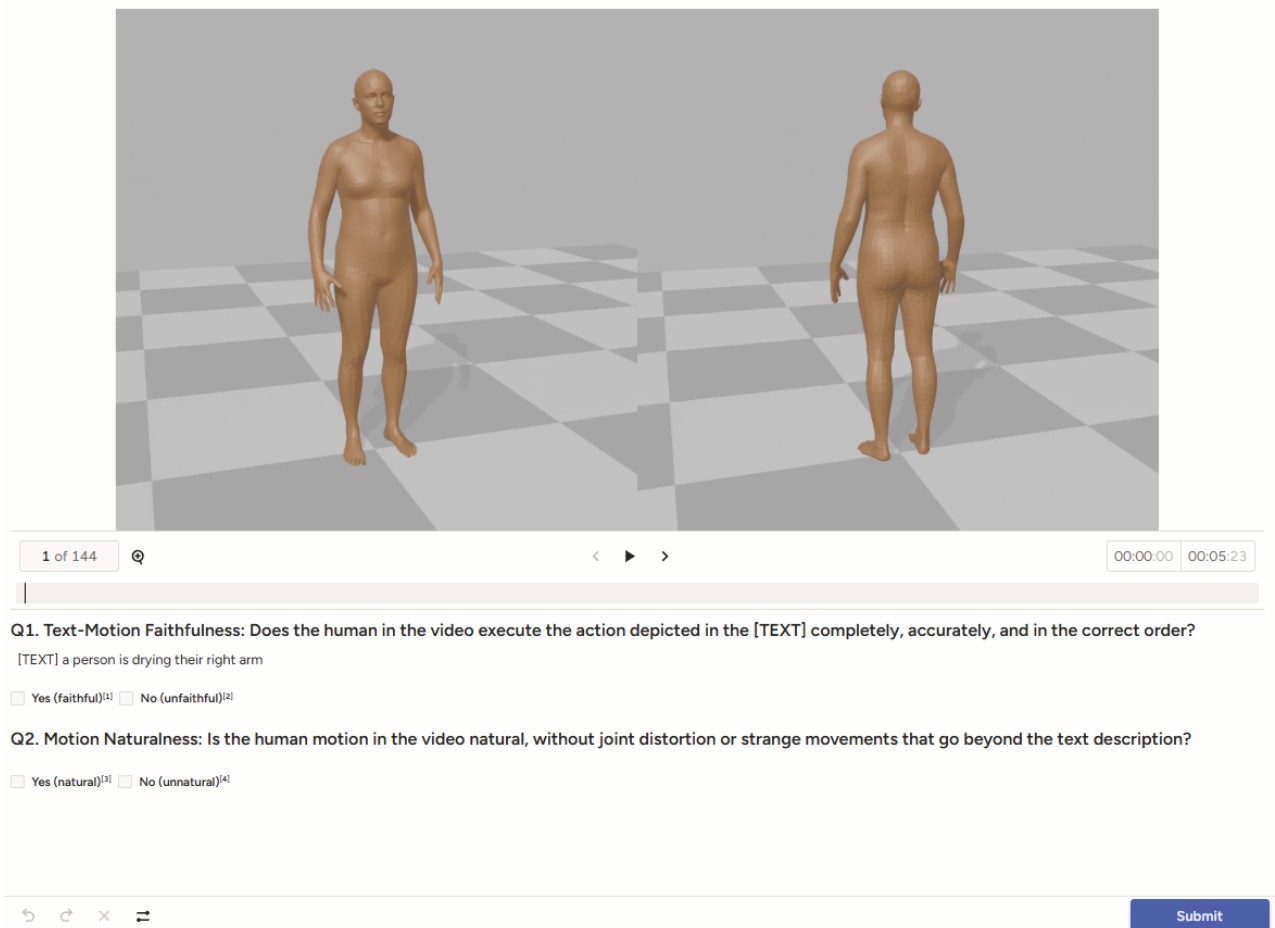

*Figure 5.* The user-interface of LabelStudio used for human annotation.

quality and file size, resulting in a smooth motion video that clearly presents the details of the generated motion sequence. Finally, we use LabelStudio as a frontend for human annotation, and the user-interface is shown in Figure 5.

### A.3. Experiments on Model-Level Correlation

**Settings.** For the data belonging to the MDM split, we randomly divide it into two groups and repeat this process 10 times, resulting in a total of 20 random sub-splits. We perform the same procedure for the MotionGPT split, which also generates 20 random sub-splits. Ultimately, we obtain 40 sub-splits in total, with each sub-split containing approximately 500 data samples. Subsequently, each sub-split is analyzed independently: first, we calculate the FID score following (Tevet et al., 2022); then, we sum up all text-motion scores for each evaluation method. Finally, for each metric, we have a sequence of model-level scores with a length of 40, which can be used to compute the correlation coefficients $\tau$ and $\rho$. Table 10 reports model-level correlation of different evaluation methods.

**Based on Table 10, we have several findings:** 1) First, we observe that the model-level MoBERT score of the model fine-tuned on human-labeled samples exhibits a negative correlation with human scores, which is contrary to the case at the sample level (see Table 3). This may be attributed to the insufficient amount of human-labeled data, which causes the fine-tuned model to suffer from overfitting and assign extreme scores for false positive/negative samples. 2) Second, among automatic methods, VeMo ranks first in terms of model-level correlation, followed by reference-based methods. This is consistent with the findings from the sample-level evaluation (Table 3), indicating that both VeMo and reference-based methods can provide stable and consistent evaluations. 3) Finally, VeMo's model-level scores achieve or even surpass human user scores. The reason for this is that users are instructed to make binary selections, and inconsistent cases largely affect model-level scores. In contrast, VeMo can generate intermediate scores for cases it is uncertain about, thereby improving

*Table 10.* Model-level correlation between different evaluation scores and human judgements, reported on meta-evaluation dataset. ↑ means larger is better.

| Evaluation Method | $\tau$ ↑ | $\rho$ ↑ | Evaluation Method | $\tau$ ↑ | $\rho$ ↑ |
|---|---|---|---|---|---|
| *(reference-based automatic evaluation method)* | | | | | |
| Minus L1 Distance (w/ ref.) | 0.6138 | 0.8172 | Minus FID (w/ ref.) | 0.6032 | 0.8066 |
| *(reference-free automatic evaluation method)* | | | | | |
| Minus Multimodal Distance | 0.5503 | 0.7600 | MoBERT-base | 0.4762 | 0.7043 |
| R@1-Precision | 0.5745 | 0.7922 | MoBERT-F | -0.4021 | -0.5779 |
| R@2-Precision | 0.5676 | 0.7595 | MoBERT-N | -0.4127 | -0.6622 |
| R@3-Precision | 0.6096 | 0.7988 | MoBERT-max(F/N) | -0.4868 | -0.7193 |
| MotionCritic | 0.5609 | 0.7901 | MoBERT-min(F/N) | -0.4339 | -0.6110 |
| **VeMo (min entropy view)** | 0.7196 | 0.8774 | **VeMo (human-opt view)** | 0.7090 | 0.8834 |
| User-1 Score | 0.6915 | 0.8758 | User-2 Score | 0.5532 | 0.7726 |

*Table 11.* Usefulness of Eq. (1).

| VeMo, human-opt view | AUC-ROC ↑ | AUPR ↑ | KS ↑ | $\tau$ ↑ | $\rho$ ↑ |
|---|---|---|---|---|---|
| InternVL3-14B (w/ Eq. (1)) | **0.723±.000** | 0.740±.000 | **0.342±.000** | **0.315±.000** | **0.385±.000** |
| InternVL3-14B (w/o Eq. (1)) | 0.627±.006 | **0.743±.004** | 0.254±.012 | 0.262±.013 | 0.262±.013 |

its model-level performance. Overall, although model-level scores cannot be used to evaluate text-motion alignment, the experimental results demonstrate that VeMo has the potential to evaluate the overall performance of T2M models.

### A.4. Extra Analysis

**What about using VLM's output sentence instead of the predicted distribution?** To study the usefulness of Eq. (1), we replace Eq. (1) with matches of "yes" or "no" from the sentences generated by the VLM. This produces binary (i.e., 0/1) prediction scores, then we compare these scores with our original distribution-based scores. Note that the entropy of the output sentence of the VLM cannot be calculated, so we evaluate it under the same setting of human-opt view as in Table 4. The results in Table 11 show that VeMo w/ Eq. (1) outperforms VeMo w/o Eq. (1) in terms of AUC, KS, $\tau$ and $\rho$ (most important), indicating that Eq. (1) can significantly enhance the correlation between the evaluation scores and humans. VeMo w/o Eq. (1) yields binary scores, and the precision and recall under most thresholds are the same, which is slightly beneficial for AUPR calculation. Notably, the VeMo scoring procedure (soft distributional scoring per Eq. (1)) yields stable, zero-variance scores for deterministic input video. The VeMo scores are substantially more reliable than scores obtained via naive deterministic decoding of VLM outputs (i.e., w/o Eq. (1)).

**Computational overhead and efficiency tradeoff of the VeMo.** Our full VeMo pipeline consists of three stages: Joint-to-Mesh, Mesh-to-Video, and VLM inference. Converting a rendered 3D mesh into multi-view videos does not increase the Joint-to-Mesh overhead. For the rendering process, we measured the time and peak RAM using the Rendering-to-Body-Model video converter: the per-motion time and peak RAM for the Joint-to-Mesh stage are 182s and 793MiB, respectively, while the corresponding values for the Mesh-to-Video stage are 31s and 256MiB. The runtime, memory footprint, and evaluation performance of VeMo under different VLM configurations (human-opt view) are shown in Table 12. The primary time bottleneck is rendering, but rendering is highly parallelizable because its peak RAM is low. The primary RAM bottleneck is VLM inference; using smaller VLMs (for example, InternVL3-1B) reduces memory requirements at the cost of modest performance degradation. This trade-off makes VeMo practical for different resource budgets.

*Table 12.* VeMo Performance Under Different VLM Configurations (Human-Opt View).

| VeMo (human-opt view) | num frames | Per-Video Time/Peak-RAM | AUC-ROC | AUPR | KS | $\tau$ | $\rho$ |
|---|---|---|---|---|---|---|---|
| InternVL3-14B | 32 | 1.597s / 33221MiB | 0.723 | 0.740 | 0.342 | 0.315 | 0.385 |
| InternVL3-14B | 8 | 0.415s / 30000MiB | 0.720 | 0.738 | 0.343 | 0.311 | 0.381 |
| InternVL3-8B | 32 | 0.889s / 18332MiB | 0.687 | 0.706 | 0.291 | 0.264 | 0.324 |
| InternVL3-8B | 8 | 0.233s / 15998MiB | 0.683 | 0.701 | 0.284 | 0.258 | 0.316 |
| InternVL3-1B | 32 | 0.295s / 4470MiB | 0.630 | 0.627 | 0.215 | 0.184 | 0.225 |
| InternVL3-1B | 8 | 0.084s / 2503MiB | 0.642 | 0.645 | 0.231 | 0.201 | 0.246 |

**Efficient version of VeMo.** Distilling VLM into a smaller scoring model is attractive for efficiency. However, T2M generation exhibits many diverse, valid solutions and limited coverage of motion space; a distilled model risks overfitting to limited T2M data in much the same way as current reference-free evaluators. To address the practical time bottleneck from mesh rendering, we explored a lightweight alternative: directly visualizing joint trajectories as stick-figure videos (i.e., skipping Joint-to-Mesh). This eliminates the rendering overhead while preserving temporal joint information for the VLM. The runtime and memory profile for this converter is negligible: the Visualizing-to-Stick-Figure-Video converter achieves a per-frame time of 0.007s with a peak RAM of 0MiB. As shown in Table 13, we evaluated VeMo using stick-figure videos. Although absolute performance drops relative to full-body renderings, VeMo on stick figures still substantially outperforms the best reference-free baseline and is therefore suitable for rapid, online analyses and iterative workflows.

*Table 13.* VeMo performance on stick-figure videos.

| VeMo on Stick-Figure-Video | AUC-ROC | AUPR | KS | $\tau$ | $\rho$ |
|---|---|---|---|---|---|
| InternVL3-14B (32-frame) | 0.608 | 0.626 | 0.145 | 0.153 | 0.187 |
| InternVL3-14B (8-frame) | 0.619 | 0.633 | 0.176 | 0.169 | 0.206 |
| InternVL3-8B (32-frame) | 0.604 | 0.607 | 0.148 | 0.147 | 0.180 |
| InternVL3-8B (8-frame) | 0.606 | 0.608 | 0.159 | 0.150 | 0.184 |
| InternVL3-1B (32-frame) | 0.560 | 0.548 | 0.108 | 0.085 | 0.104 |
| InternVL3-1B (8-frame) | 0.571 | 0.563 | 0.118 | 0.100 | 0.123 |

**Multi-view fusion.** Since different camera views may reveal complementary aspects of a 3D motion, we further examine whether current VLMs can directly integrate multi-view visual evidence. Using InternVL3-14B with 32 input frames, we compare the human-opt view with two naive fusion strategies: *multi-view per frame*, where each frame contains two synchronized views, and *multi-view in sequence*, where multiple views are arranged along the temporal input sequence. The human-opt view achieves an AUC-ROC of 0.723, AUPR of 0.740, KS of 0.342, Kendall's $\tau$ of 0.315, and Spearman's $\rho$ of 0.385. In comparison, multi-view per frame obtains 0.716, 0.740, 0.348, 0.306, and 0.374, while multi-view in sequence obtains 0.711, 0.728, 0.328, 0.299, and 0.366, respectively. These results indicate that simply exposing the VLM to more views does not improve VeMo and can slightly degrade rank correlation, likely because current VLMs are not yet optimized for temporally consistent multi-view motion reasoning. We therefore treat multi-view integration as a promising but nontrivial direction for future work, rather than assuming that simple visual aggregation is sufficient.

## A.5. Future Direction.

**Evaluation on failure severity.** VeMo is designed to answer the most fundamental measurable question in T2M evaluation: whether the generated motion matches the text. Since existing automatic metrics still struggle with coarse-grained "yes/no" alignment, we focus on this well-defined setting. Assessing finer degrees of failure, such as minor mistakes versus complete mismatch, requires richer annotations and more capable evaluators, and remains an important direction for future work.

**Scaling the meta-evaluation.** The meta-evaluation benchmark introduced in this work has a focused goal: to construct a fair human-rating reference and to verify the basic evaluation effectiveness of T2M evaluators. Our experiments and analyses are framed around this objective, and they demonstrate that the VLM-based T2M score is substantially closer to human ratings than existing automatic evaluators, which is an important step toward improving T2M evaluation. Notably, scaling the meta-evaluation to cover a much wider range of motion complexity and more T2M models is valuable in the future but also labor-intensive and outside the core contributions claimed in this paper.

**Boundaries of entropy-based view selection.** Entropy-based view selection is not a guarantee of correctness: VLMs may still make low-entropy but incorrect predictions on out-of-distribution videos (Liu et al., 2024; Farquhar et al., 2024). This reflects the gap between current VLM video understanding and human motion perception, and suggests the need for better motion grounding. Moreover, perfect agreement with human raters is inherently unattainable (Table 3), since T2M alignment is subjective and multimodal. Our goal is thus to narrow the gap between automatic evaluators and human judgment.

**Beyond the offline analysis presented**, we believe VeMo can provide concrete value for T2M training in two practical ways: (1) Training-data curation. Recent efforts augment training sets with motions recovered from video (Ding et al., 2025). VeMo can serve as an automatic filter to remove low-quality or noisy converted motions from the training set. (2) Training-time reward shaping. VeMo can be combined with offline reinforcement-learning schemes as an auxiliary reward signal; by periodically re-scoring and updating offline data, models can be steered toward higher-quality generations.

### A.6. Limitations and Broader Discussion

VeMo provides a zero-shot evaluator for text-motion alignment, but it has several limitations. First, the current benchmark is constructed from HumanML3D prompts and four T2M generators. Although the added StableMoFusion and MotionLCM-V2 split strengthens external validation, broader coverage of motion representations, datasets, and generation paradigms remains future work. Second, VeMo depends on the visual proxy produced by rendering. Camera placement, body appearance, lighting, and visual artifacts can affect VLM judgments, and severe rendering noise may lead to confidently incorrect low-entropy predictions. Third, VeMo inherits the semantic biases and perception limits of the underlying VLM (Fei et al., 2023). Improvements in video-language understanding are likely to improve VeMo. Fourth, full-body rendering introduces computational overhead, although the stick-figure variant provides a faster alternative for large-scale or iterative use. Finally, this work focuses on coarse-grained text-motion alignment. Fine-grained diagnostic T2M evaluation, such as identifying which body part or temporal segment causes a mismatch, is an important direction for future research.

