# OpenReview forum: "Zero-Shot Text-to-Motion Evaluation using Video Language Models"
_ICML.cc/2026/Conference — ICML 2026 regular_

### Official Review · Reviewer_XHxs · 2026-03-09

**Soundness:** 2
**Presentation:** 3
**Significance:** 3
**Originality:** 3
**Overall Recommendation:** 4
**Confidence:** 2

**Summary:**

This paper proposes VeMo, a zero-shot evaluation framework for Text-to-Motion (T2M) models. To avoid over-relying on ground-truth motion data, the method renders generated 3D motions into 2D videos and leverages a Video Language Model (VLM) for scoring. To address occlusion issues during the 3D-to-2D projection, it introduces an entropy-based uncertainty analysis to automatically select the optimal viewpoint. Additionally, the paper contributes a high-quality meta-evaluation benchmark comprising 2,202 annotated pairs.

**Compliance With Llm Reviewing Policy:**

Affirmed.

**Final Justification:**

My primary concerns regarding this paper were the high computational cost and the current limitation to binary alignment classification. The rebuttal has addressed my questions and appropriately contextualized the limitations of the proposed framework. I believe the strengths now outweigh the weaknesses.

**Key Questions For Authors:**

1. Given the substantial time overhead introduced by the 3D-to-2D rendering process (e.g., 182s per motion), how do you envision VeMo being practically applied during the training phase of T2M models, where rapid, large-scale feedback is required?
2. VeMo currently focuses on binary alignment classification. Do you have any preliminary ideas or plans to extend this framework to provide fine-grained evaluations (e.g., assessing the speed, amplitude, or the accuracy of a specific body part)?
3. The appendix mentions that directly concatenating multiple views in a single frame confuses the VLM. Have you experimented with inputting multiple views as "sequential video frames," or utilizing VLMs with native multi-image input capabilities to improve performance?

**Limitations:**

yes

**Strengths And Weaknesses:**

Strength:
1. Originality: Framing 3D motion evaluation as a 2D VLM video reasoning task, and using "entropy" to mitigate viewpoint occlusion, is a highly novel approach.
2. Significance: The lack of reliable evaluation metrics is a major bottleneck in the T2M field. Providing a data-independent evaluation method and an open-source, high-quality testing benchmark is a significant contribution to the community.

Weakness:
1. The evaluation cost is quite high. The 3D-to-2D rendering process (Joint-to-Mesh) is very time-consuming (e.g., 182 seconds per motion), making it difficult to use for large-scale, rapid iterative evaluations during model training.
2. Currently, the method primarily outputs coarse-grained binary "yes/no" alignment scores. It lacks the ability to provide fine-grained diagnostic feedback for minor motion details (e.g., the exact bending of a specific joint or the precise speed of the action).

---

> ### Author Rebuttal · Authors · 2026-03-30
>
> We sincerely thank Reviewer for recognizing the originality of framing 3D motion evaluation as a 2D VLM reasoning task and the significance of providing a data-independent evaluation benchmark.
>
> >Further Training Application: The reviewer has raised an insightful question. This work focuses on analyzing and improving existing T2M evaluation using VLMs, yet the limitations regarding future practical applications also require sufficient discussion. Although VeMo offers substantial efficiency gains over human evaluation, its inference speed still cannot meet the requirements of online training. At present, VeMo can only potentially assist training in the context of offline reinforcement learning algorithms. We will emphasize this point in the Limitations section to outline clear directions for future research.
>
> >Preliminary Ideas on Fine-Grained Evaluations: While we do not yet have a mature solution for fine-grained evaluation, we have preliminary ideas in this direction. Given that we have shown VLMs can assign reliable overall scores to motion clips, a natural next step is to attribute these scores to individual frames and quantify the contribution of each frame. Such frame-level signals could also serve as process rewards for training T2M models, avoiding the waste of high-quality frames during rollouts. We hope our analytical work can lay a foundation for broader research across the field.
>
> >Further Multi-View Results: For the multi-view experiment in Appendix A.4.4, we originally concatenated multiple perspectives within each frame of the input video, and observed a slight performance drop. Following the reviewer’s suggestion, we also sequentially concatenated videos from two views (one human-optimal view and one randomly rotated view) under the same experimental setup as A.4.4, constructing a longer sequential video as input to VeMo. The complete results are as follows:
> >
> > | Ablation on View | AUC-ROC↑ | AUPR↑  | KS↑    | Kendall ↑ | Spearman ↑ |
> > |---------------------------------|---------|-------|-------|--------------|---------------|
> > |  human-opt view             | 0.723   |  0.740 |  0.342 |  0.315       |  0.385         |
> > |  multi-view per frame      | 0.716   |  0.740 | 0.348 |  0.306        |  0.374         |
> > |  multi-view in sequence  | 0.711   |  0.728 | 0.328 |  0.299        |  0.366         |
>
>
> >We agree with the reviewer’s suggestion to integrate VLMs with native multi-image input capabilities to further improve performance. In Table 4, we distinguish three types of model architectures, among which the InternVL3 series represents VLMs with native multi-image reasoning ability. We will highlight this comparison more prominently in the revision.

---

> > ### Author Rebuttal · Reviewer_XHxs · 2026-04-03
> >
> > The rebuttal has adequately addressed my questions, and I will raise my score to reflect this.

---

> > > ### Author Response · Authors · 2026-04-03
> > >
> > > Thank you for taking the time to review our work. We are glad to have addressed your concerns.

---

### Official Review · Reviewer_qUPw · 2026-03-11

**Soundness:** 2
**Presentation:** 3
**Significance:** 3
**Originality:** 3
**Overall Recommendation:** 4
**Confidence:** 4

**Summary:**

This paper proposes VeMo, a zero-shot evaluation framework for text-to-motion (T2M). The main idea is to render generated 3D motions into skinned videos and use a video-language model (VLM) to judge text-motion alignment. To reduce information loss from 3D-to-2D projection, the method selects the view with the lowest predictive entropy. The paper also introduces a meta-evaluation benchmark built from 1,101 HumanML3D test prompts and 2,202 generated text-motion pairs from MDM and MotionGPT, with human annotations for Faithfulness, Naturalness, and aggregated Alignment. Experiments show that VeMo correlates better with human judgments than several existing reference-based and reference-free baselines.

**Compliance With Llm Reviewing Policy:**

Affirmed.

**Final Justification:**

The rebuttal addresses my main concerns sufficiently. In particular, the authors clarified the extent of regeneration and provided additional results on new generator outputs without regeneration, where VeMo still leads prior baselines. Some robustness and generalization concerns remain, but overall I now view the paper as borderline accept rather than borderline reject.

**Key Questions For Authors:**

1. Please clarify the “regenerate controversial motions for re-annotation until full consensus is reached” step. How many samples were regenerated, and do the main conclusions still hold if only the original generated motions are used? This would significantly affect my assessment of the benchmark’s soundness.
2. Can the authors provide stronger out-of-distribution validation, e.g., more T2M generators, different training paradigms, or different motion representations/datasets? This would strengthen the generalization claims substantially.
3. How sensitive is VeMo to the instruction template, yes/no label wording, rendering setup, camera placement, lighting, and body appearance? A robustness analysis would help separate the method’s core idea from pipeline-specific engineering choices.
4. What is the main practical advantage of entropy-based view selection over a well-designed fixed view, given that Table 4 shows only modest differences?

**Limitations:**

yes, but the main paper should more directly acknowledge the restricted benchmark scope, possible sensitivity to rendering/prompt choices, VLM bias, and the substantial computational overhead

**Strengths And Weaknesses:**

Strengths:
1. The paper studies an important problem, since reliable evaluation remains a major bottleneck for progress in text-to-motion generation. The core idea of converting motion evaluation into a video-based reasoning problem is intuitive and reasonably novel in this context.
2.  The benchmark contribution is also valuable: a test-only meta-evaluation dataset with both coarse and fine-grained annotations is useful for comparing evaluators more fairly. Empirically, VeMo shows clear gains over prior reference-free baselines on the proposed benchmark, and the paper includes useful ablations on VLM backbones, frame counts, and view selection.

Weaknesses:
1. The main concern is external validity. The benchmark is built from HumanML3D prompts and only two generators, MDM and MotionGPT, so the evidence is not yet strong enough to support broad claims about generality across motion representations or T2M settings.
2. The benchmark construction protocol: the paper states that controversial cases may be regenerated and re-annotated until full consensus is reached. This may alter the original sample distribution and potentially make the benchmark easier, so the authors should quantify how often this happened and show that the conclusions hold without this intervention.
3. The entropy-based view selection is interesting but somewhat overstated: in Table 4, the human-optimized view is already as good as or slightly better than min-entropy selection on some ranking metrics, so this component currently looks more like a practical heuristic than a strongly validated core contribution. 4. The method may be sensitive to rendering choices and prompt design, but robustness to these choices is not studied thoroughly enough.

Overall, I think the paper has clear merit and presents a promising direction, but the current empirical evidence is not yet strong enough for acceptance without revision.

---

> ### Author Rebuttal · Authors · 2026-03-30
>
> We sincerely thank the reviewer for acknowledging the importance of addressing text-to-motion evaluation, the novelty of our VLM-based approach, and the value of introducing a test-only meta-evaluation benchmark. We address your concerns below:
>
> >Regeneration of Controversial Samples and More OOD Validation: In the original submission, regeneration occurred in ~12% of cases, primarily ambiguous instances where normal motion patterns interspersed with abnormal movements led to interpretive disputes. We have expanded the benchmark with outputs from StableMoFusion and MotionLCM‑V2 without regeneration, and this means the expanded dataset contains more ambiguous cases. The results on the OOD split (StableMoFusion & MotionLCM‑V2) are as follows:
> >
> > | Evaluation on New Splits (StableMoFusion & MotionLCM-V2) | AUC-ROC↑ | AUPR↑  | KS↑    | Kendall ↑ | Spearman ↑ | p_value ↓              |
> > |---------------------------------|---------|-------|-------|--------------|---------------|-----------------------|
> > | Minus L1 Distance (reference-based)               | 0.568   | 0.696 | 0.112 | 0.091        | 0.112         | 7.864718680510577e-08 |
> > | Minus Multimodal Distance       | 0.519   | 0.658 | 0.046 | 0.026        | 0.032         | 0.06588550277403615   |
> > | R1-Precision                    | 0.512   | 0.723 | 0.024 | 0.023        | 0.023         | 0.13592633444667507   |
> > | R2-Precision                    | 0.514   | 0.752 | 0.028 | 0.027        | 0.027         | 0.10544462113766229   |
> > | R3-Precision                    | 0.516   | 0.773 | 0.032 | 0.032        | 0.032         | 0.06768722050001316   |
> > | VeMo (human-opt view)           | 0.621   | 0.765 | 0.212 | 0.163        | 0.200           | 3.777051764000001e-21 |
> > | VeMo (min entropy view)         | 0.625   | 0.77  | 0.226 | 0.169        | 0.207         | 1.2242719400000001e-22|
> >
> > The overall evaluation performance of the baseline metrics has declined, but VeMo still maintains a clear lead.
>
> >Regarding prompt sensitivity: We emphasize that the core mechanism of VeMo relies on the zero-shot video reasoning and text-understanding capabilities inherent to state-of-the-art VLMs. Text understanding is a primary design focus of these foundation models. Consequently, VeMo's performance depends on whether the VLM can fully comprehend text and visual semantics; complex or heavily engineered prompting is not a direct factor in improving this baseline alignment. We will strengthen the discussion of this reliance on VLM text understanding in the revision.
> >
> >We agree that understanding the impact of rendering is crucial. As verified in Section A.4.3, minimalist stick-figure rendering degrades model performance, and rendering-related factors alongside VLM constraints represent the main limitations of VeMo (as discussed in Appendices A.5 and A.6).
>
> >Advantage of Entropy-based Selection: This is a reasonable concern. We do not view our method as replacing a well-designed fixed view. Instead, it automates view selection in settings where a human-designed camera setup is unavailable. The human-optimized view serves as an oracle reference, and minimum-entropy selection provides a practical approximation thereof. The modest performance gap in Table 4 is expected given the strong performance of the oracle view; we will revise our wording to clarify this distinction.
>
> >Limitations: We agree that the current limitation discussions are relatively scattered across A.5 and A.6 and lack comprehensiveness. In the revised paper, we will add a dedicated limitations section to more explicitly acknowledge the restricted scope of the current benchmark models, the inherent semantic biases of the underlying VLM, and the computational overhead introduced by the rendering process (Table A.10).

---

> > ### Author Rebuttal · Reviewer_qUPw · 2026-04-03
> >
> > The rebuttal addresses my main concerns sufficiently. In particular, the authors clarified the extent of regeneration and provided additional results on new generator outputs without regeneration, where VeMo still leads prior baselines. Some robustness and generalization concerns remain, but overall I now view the paper as borderline accept rather than borderline reject.

---

> > > ### Author Response · Authors · 2026-04-04
> > >
> > > Thank you for taking the time to review our work. We are glad to have addressed your main concerns.

---

### Official Review · Reviewer_sQ9T · 2026-03-11

**Soundness:** 3
**Presentation:** 3
**Significance:** 3
**Originality:** 2
**Overall Recommendation:** 4
**Confidence:** 3

**Summary:**

This paper proposes VeMo, a zero-shot evaluation framework for text-to-motion generation that renders generated 3D motions into videos and leverages video-language models to assess text–motion alignment. It also introduces a human-annotated meta-evaluation benchmark for evaluating T2M metrics and shows that VeMo achieves stronger correlation with human judgments than existing reference-based and reference-free evaluation methods.

**Compliance With Llm Reviewing Policy:**

Affirmed.

**Final Justification:**

The rebuttal addresses my main concerns.

**Key Questions For Authors:**

1. The meta-evaluation benchmark only contains motions generated by MDM and MotionGPT. How well does VeMo generalize to outputs from other T2M models with different motion characteristics?
2. The method selects the view with the minimum predictive entropy as the evaluation input. Are there cases where the lowest-entropy view produces a confidently incorrect evaluation?
3. Since the method relies on rendering motion into video, how sensitive are the results to rendering parameters such as camera distance, lighting, body scale, or background?
4. The evaluation uses a binary yes/no likelihood formulation. Did the authors explore alternative prompting strategies (e.g., multi-step reasoning or textual explanation-based scoring)? It would be interesting to see whether richer prompting could further improve correlation with human judgments.

**Limitations:**

Yes

**Strengths And Weaknesses:**

Strengths:
- The motivation, limitations of current text-to-motion evaluation metrics, is clearly explained, and the overall pipeline is straightforward.
- By leveraging large-scale pretrained video-language models, the proposed approach introduces a potentially scalable direction for automatic evaluation that does not require motion-specific data.
- The formulation of evaluation as a zero-shot video reasoning task and the entropy-based view selection for mitigating rendering ambiguity are interesting contributions.

Weaknesses:
- Although the experimental results show clear improvements over prior automatic metrics, the evaluation is limited to motions generated by only two T2M models (MDM and MotionGPT). It remains unclear how well the proposed metric generalizes to outputs from a broader range of generation methods or datasets.

- The scale of the meta-evaluation dataset (2202 text-motion pairs) is relatively modest. This may limit the statistical robustness of the conclusions and the generality of the benchmark.

- The method depends on relatively large video-language models, which may introduce computational overhead and limit practical applicability for large-scale evaluation pipelines.

---

> ### Author Rebuttal · Authors · 2026-03-30
>
> We thank the reviewer for the positive assessment of our scalable research direction and the formulation of our zero-shot reasoning task.
>
> >Benchmark Scale and Generalization: To address the concern regarding limited motion generators, we have expanded the meta-evaluation benchmark by including 1,101 motion samples each from StableMoFusion and MotionLCM-V2. This doubles our dataset size and incorporates modern diffusion-based and consistency models, demonstrating VeMo's robustness across diverse motion characteristics. Please refer to the response to Reviewer qUPw for the experimental results.
>
>
> >Corner Case Discussion: We have analyzed this issue in Figure 3 and Section 4.3 of the paper. When rendered videos contain severe noise, the lowest-entropy view can yield confidently incorrect evaluations. We will highlight this limitation in the revised manuscript.
>
> >Sensitivity to Rendering Parameters: The stick-figure experiment in the appendix analyzes the impact of minimal rendering settings (skeleton only, no lighting/background/skin, farther camera). As discussed in Appendices A.5 and A.6, both VLMs and rendering choices influence VeMo's evaluation performance, which also constitutes one of the limitations of our method. We summarize several key results as follows; the complete results can be found in A.Table 11.
> >
> > | Methods | AUC-ROC↑ | AUPR↑  | KS↑    | Kendall ↑ | Spearman ↑ |
> > |---------------------------------|---------|-------|-------|--------------|---------------|
> > | InternVL3-14B (32-frame, default rendering)     | 0.723   | 0.740 | 0.342 |  0.315  |  0.385  |
> > | InternVL3-14B (32-frame, stick-figure rendering)     | 0.608  | 0.626 |  0.145 | 0.153 | 0.187 |
> >
> >Golden rendering protocol is hard to define. VeMo is actually imitating the human annotation pipeline: humans select preferred rendering settings to visualize motions and then manually evaluate data, and we have replaced the manual part of this pipeline with VLMs. Once VLMs achieve human-level multimodal understanding, the VeMo pipeline is expected to achieve human-level motion evaluation.
>
> >Richer prompting strategies and multi-step reasoning represent promising directions for future work, the introduction of VLM into T2M evaluation enables LLM-based inference augmentation to be directly implemented in this field. In addition to prompt engineering, relevant expansion directions include multi-granularity evaluation, score attribution, RAG, etc. We will strengthen discussions on these in future work.

---

> > ### Author Rebuttal · Reviewer_sQ9T · 2026-04-03
> >
> > While I appreciate the additional experiments and clarifications provided in the rebuttal, several core concerns remain only partially addressed. In particular, the entropy-based selection mechanism is shown to fail under certain conditions without proposing a possible mitigation strategy, and key design aspects such as prompting strategies are not explored under limited-sample settings.

---

> > > ### Author Response · Authors · 2026-04-03
> > >
> > > We sincerely apologize for not fully resolving your concerns in our initial response, which led to additional effort on your part to follow up with further questions. We have carefully addressed all your concerns to the best of our ability and would greatly appreciate it if you could reconsider evaluating our work once these doubts are resolved.
> > >
> > > > Our proposed mitigation strategy is detailed in Sec. 3.2.1 (line 120) and Sec. 4.3 (line 249). Our analysis shows the entropy-based selection mechanism fails when video motion information falls below a certain threshold (Sec. 4.3, Fig. 3). Based on the visibility ablation (Fig. 3), we introduce a simple yet effective fix using whole body visibility as a core rendering constraint. The VeMo pipeline outperforms the baseline metrics (Tables 3 and 5), also demonstrating the effectiveness of this strategy.
> > >
> > >
> > > > This work focuses on analyzing the reliability of VLM as a zeroshot T2M evaluator, and the advantages and limitations of entropy-based selection and VLM scoring have been carefully analyzed both theoretically and empirically in this paper. We propose the mitigation method only for the sake of completeness, so it is not included in the core contributions of the paper. We hope our analytical work can lay a foundation for broader research across the field. We will have a more explicit discussion in the revision.
> > >
> > > > We are pleased to address the reviewer's concerns regarding prompting strategies. However, it should first be clarified that prompting strategies are not a key design of the VeMo pipeline, and no prompt optimization was performed on any VLMs. We first drafted instructions to collect meta-evaluation data and then used these instructions on VLMs. We tested a wide range of VLMs (Tab. 4) using the same prompts, and the experimental results have succinctly reflected the robustness. We do not wish to over-optimize the prompts; if VLMs can achieve human-level VL understanding in the future, they are expected to handle any human-centric prompts.
> > >
> > > > We believe the ablation experiments related to prompting strategies are quite interesting and are currently being supplemented. The experimental setup is as follows: before inputting prompts into VLMs, we use GPT-4o-mini to randomly paraphrase the prompts. The experimental results demonstrate the extent to which VeMo is affected by prompting strategies, and it still maintains better correlation with human evaluation compared with the baselines metrics.
> > > >
> > > >| Ablation on Prompt	| AUC-ROC↑	| AUPR↑	|	KS↑	| Kendall ↑	| Spearman ↑   |
> > > >|:--------------------------|---------------:|---------------:|---------------:|---------------:|---------------:|
> > > >| default prompt  (Fig. 2 in paper, Tab. 4 results) |	0.723 |	0.740 |	0.342 |	0.315 |	0.385 |
> > > >| default prompt paraphrased by GPT-4o-mini  |	0.708 |	0.731 |	0.322 |	0.294 |	0.360 |
> > > >| best reference-free baseline (Tab. 3 results) |	0.549 | 0.553 | 0.088 | 0.069 | 0.084  |
> > > >
> > > > **Prompt used for paraphrasing:**
> > > >```
> > > ><original_prompt>
> > > >{raw_input_text}
> > > ></original_prompt>
> > > >Please rephrase all of the texts in the <original_prompt> tags into another expression with the same meaning.
> > > >
> > > >The rephrased results should be placed within the <rephrased> tags, and the format is as follows:
> > > ><rephrased>
> > > >$rephrased_expression
> > > ></rephrased>
> > > >
> > > >Please directly return the <rephrased> tags and the content inside them without any extra explanation:
> > > >```

---

### Official Review · Reviewer_ToSD · 2026-03-12

**Soundness:** 2
**Presentation:** 3
**Significance:** 2
**Originality:** 3
**Overall Recommendation:** 4
**Confidence:** 4

**Summary:**

In this paper, the authors propose VeMo, a reward function for text-to-motion generation.
VeMo is based on a VLM prompted with rendered views of the generated motion along with two binary questions regarding motion–text alignment and motion naturalness. The final score is computed by aggregating the output likelihoods produced by the VLM.
To mitigate the influence of the chosen rendering viewpoint, the authors propose prompting the VLM with K different views of the same motion and selecting the output with the lowest entropy.
Finally, they introduce a curated evaluation benchmark containing human-annotated motion–text pairs.

**Compliance With Llm Reviewing Policy:**

Affirmed.

**Final Justification:**

The rebuttal has globally addressed my concerns, notably regarding the missing related work and qualitative results. I am leaning toward acceptance; however, my decision remains borderline, as the entropy-based selection vs human-opt setup, and time complexity makes the overall message of the paper insufficiently clear to me. Nevertheless, I acknowledge the quality of the work and thank the authors for their rebuttal.

**Key Questions For Authors:**

- Is the entropy-based output selection truly useful if the final configuration uses only one view with the “human-opt” setup?

**Limitations:**

yes

**Strengths And Weaknesses:**

## Strengths

+ The entropy-based output selection and the zero-shot nature of the method are interesting.
+ The ablation study is well conducted and helps clarify the contribution of each component.
+ The paper is well written and easy to follow.

## Weaknesses

- The authors omit a relevant published related work: PP-Motion (Zhao et al., ACM Multimedia 2025). Although the code is not open-sourced, the work should at least be discussed in the related work section. Moreover, the authors could still compare their approach by evaluating it on the publicly available MotionPercept dataset from MotionCritic (Wang et al., ICLR 2025), for which PP-Motion results are reported.
- It is unclear what the final proposed method is. In Section 5.4 (“Impact of K”), the authors conclude that the best setup is the “human-opt” configuration using only one view, which makes the entropy-based output selection useless.
- The time complexity for scoring a single motion appears high. The authors report approximately 30 seconds per view, and the optimal number of views is claimed to be two. This results in at least one minute per sample, which may be impractical for large-scale evaluation. Also, it is not clear in Table 10 if the reported times are only the VLM inferences or the total pipeline.
- The paper lacks qualitative results.

Minor concerns:
- Other open-source VLMs are available (e.g., Qwen-VL, LLaMA Vision, and DeepSeek-VL). Including them would strengthen the ablation study in Table 4 and help better understand the influence of the underlying VLM.
- It would be interesting to analyze the impact of the rendering process (e.g., resolution, realistic rendering vs. the current setup, etc.).
- It would also be interesting to use VeMo as a reward for fine-tuning and compare its effectiveness with MotionCritic.

---

> ### Author Rebuttal · Authors · 2026-03-30
>
> We thank the reviewer for valuable feedback and for highlighting the interest in our entropy-based method and the quality of our ablation studies.
>
> >Advantage of VeMo: VeMo is a fully automatic metric that can reliably evaluate motion without human judgement or data collection; VeMo's performance leads among existing automatic metrics; VeMo pipeline evolves with the advance of pre-trained VLMs.
>
> >Related Work: We appreciate the pointer regarding PP-Motion and will include it in the related work section. PP-Motion focuses on evaluating human motion fidelity from the perspectives of physical feasibility and MotionCritic focuses on text-independent human perceptual realism, whereas VeMo targets semantic/multimodal alignment evaluation by reasoning over rendered motion videos using VLMs. We believe these lines of work are complementary rather than overlapping. Additionally, we will extend the meta-evaluation set by adding motions generated by StableMoFusion and MotionLCM-V2, paired with human annotations collected under the same protocol. This directly addresses the concern regarding external validity while maintaining the benchmark as test-only. For detailed results, please refer to our response to Reviewer qUPw.
>
>
> >Entropy-based Selection and Human-opt: The entropy-based selection method is designed to reduce the labor costs and protocol dependency associated with manual rendering setup. While the human-opt view represents a fixed, manually engineered perspective, min-entropy selection serves as an automatic, readily deployable alternative. Table 4 already demonstrates that min-entropy selection performs consistently close to the human-opt view, and significantly better than random or max-entropy selection. We will state this comparison more explicitly in the revised manuscript to avoid ambiguity regarding the final proposed method.
>
>
> >Time Complexity: The times reported in Table 10 correspond specifically to VLM inference. The full pipeline also includes Joint-to-Mesh (182s) and Mesh-to-Video (31s), as discussed in Appendix A.4.2. While this latency is too high for real-time training, our stick-figure converter only takes 0.007s per frame to convert Joint into Video and enter the VLM inference stage. The results in Table 11 show that the pipeline combining the stick-figure converter with InternVL3-1B processes each 100-frame video in < 1 second, with affordable performance loss (still exceeding the baselines). We will clarify this distinction more clearly in the revision.
>
>
> >Qualitative Results: Figures 1 and 4 provide qualitative evidence of VeMo’s alignment with human judgment. We have also provided thousands of video demos in the supplemental materials (anonymous repo: https://anonymous.4open.science/r/VeMo-322A/README.md) to offer a broader qualitative sense of the model's performance. We will add more qualitative results in the revision.
>
> >Minor concerns: (1) We will expand the related work section to cover more open-source VLMs and discuss their impact on our method; (2) The stick-figure experiment in the appendix analyzes the impact of minimal rendering conditions. We will add more rendering analysis in revision. (3) We agree that using VeMo as a reward for fine-tuning is very promising. We will strengthen the discussion in future work; currently, we are focusing on reliable evaluation.

---

> > ### Author Rebuttal · Reviewer_ToSD · 2026-04-03
> >
> > I thank the authors for their rebuttal; however, several points remain unresolved for me.
> >
> > **Entropy-based Selection and Human-opt:** It is unclear to me to what extent the human-opt rendering setup is not automatable. The paper states: “The camera’s position is determined by offsetting the human model’s geometric center by a fixed vector (-1, -3, 0.6); similarly to the light source, the camera is rotated to face the model’s center (using the vector from the camera to the model’s center converted to an Euler angle).” Given this description, I do not understand what additional cost or manual effort is involved in the human-opt setup.
> >
> > **Qualitative results:** The videos provided in the supplementary material focus exclusively on the generated motions, which does not offer a broad qualitative assessment of the model’s overall performance.
> >
> > Minor points:
> > - The time complexity and the resulting need for a simplified version of the entropy-based selection make the narrative of the paper somewhat unclear: each component of the method is presented as a contribution, yet they are not feasible in practice and are therefore replaced by downgraded versions.
> > - As also noted by reviewer qUPw, prompt sensitivity, particularly with respect to rendering, is a very interesting direction to explore. I acknowledge that the stick-figure experiment represents a first step in this direction and effectively highlights the importance of this issue.

---

> > > ### Author Response · Authors · 2026-04-03
> > >
> > > We sincerely thank the reviewer for their continued engagement and for providing us the opportunity to further clarify our work. We apologize for any initial lack of clarity and have made every effort to address your remaining concerns comprehensively. We hope these additional details and results justify a re-evaluation of our submission.
> > >
> > > > Our human-optimized rendering setup is manually engineered in Blender: we iteratively adjust the scheme, finalize and export it only when it supports convenient, reproducible human annotation. We acknowledge the reviewer's concern that entropy-based selection may no longer deliver substantial, sustained practical benefits after the rendering setup is engineered. However, this method retains meaningful analytical value, which we emphasize in the abstract and introduction (worded as entropy-driven uncertainty analysis). Specifically, we link rendering views to VLM scores and evaluation confidence, identifying conditions where VLM evaluation is reliable or unreliable. The comparable performance of entropy-based selection to the human-optimized setup further validates this analysis. Even if the human-optimized view is more practical in application, entropy-based selection provides a principled justification for its use.
> > >
> > > > Qualitative results. VeMo scores are stored in `storage/vemo_out` in the repo, and human annotations for MotionGPT/MDM in `storage/eval_scores`. Original unsubtitled videos can be reused to reproduce evaluation scores. We have updated `README.md` to explain how to add metrics and text to videos using `src/show_metrics_on_videos.py`. For direct qualitative analysis, we have also uploaded processed video demos, refer to the link (https://anonymous.4open.science/r/VeMo-322A/README.md) for guidance. We have also added the unnormalized MotionCritic score in the qualitative case for reference only (as it is non-multimodal). We thank the reviewer for helping improve our supplementary materials.
> > >
> > > > Minor points (presentation). We will refine the presentation of our contributions. Our method is proposed and analyzed under offline, human‑intervention‑free conditions first, then human factors are considered, clarifying that manual rendering engineering provides sustained benefits. Effectiveness–efficiency trade-offs only need to be considered when discussing potential training applications. A method needs to meet different requirements under different conditions, and our original intention is to better position the method rather than overshadow the contribution.
> > >
> > >
> > > > Minor points (prompt sensitivity). We kindly refer the reviewer to reviewer qUPw for the results.

---

### Decision · Program_Chairs · 2026-04-30

**Decision:**

Accept (regular)

**Comment:**

Scores: 4 (WA), 4 (WA), 4 (WA), 4 (WA)

The paper proposes VeMo, a zero-shot text-to-motion evaluation framework that renders generated 3D motions into videos and leverages video-language models (VLMs) for scoring, with entropy-based view selection to mitigate 3D-to-2D projection issues. It also contributes a human-annotated meta-evaluation benchmark.

Reviewers acknowledge several strengths including:
- Novel framing of 3D motion evaluation as a 2D VLM video reasoning task (ToSD, sQ9T, qUPw, XHxs)
- Important and clearly motivated problem (sQ9T, qUPw, XHxs)
- Useful meta-evaluation benchmark with human annotations (qUPw, XHxs)
- Well-conducted ablation study (ToSD, qUPw)

The weaknesses noted by reviewers include:
- Limited generalization - originally only 2 T2M generators (sQ9T, qUPw)
- Ambiguous narrative around entropy-based selection vs. human-opt view (ToSD, qUPw)
- High computational cost (ToSD, sQ9T, XHxs)
- Sensitivity to rendering choices and prompts underexplored (sQ9T, qUPw)

In the rebuttal, the authors expanded the benchmark with two additional generators (StableMoFusion, MotionLCM-V2), discussed entropy-based selection, proposed a lightweight stick-figure pipeline under 1 second, and provided a prompt paraphrasing ablation.

All four reviewers gave positive recommendations post-rebuttal. The Area Chair agrees with the reviewers and recommends acceptance. Authors are advised to revise based on the discussions in the rebuttal.